# Pomegranate Extract Affects Gut Biofilm Forming Bacteria and Promotes Intestinal Mucosal Healing Regulating the Crosstalk between Epithelial Cells and Intestinal Fibroblasts

**DOI:** 10.3390/nu15071771

**Published:** 2023-04-05

**Authors:** Giulia Rizzo, Samuel Elias Pineda Chavez, Elisa Vandenkoornhuyse, Cindy Lorena Cárdenas Rincón, Valeria Cento, Valentina Garlatti, Marek Wozny, Giusy Sammarco, Alessia Di Claudio, Lisa Meanti, Sudharshan Elangovan, Andrea Romano, Giulia Roda, Laura Loy, Arianna Dal Buono, Roberto Gabbiadini, Sara Lovisa, Roberto Rusconi, Alessandro Repici, Alessandro Armuzzi, Stefania Vetrano

**Affiliations:** 1Laboratory of Gastrointestinal Immunopathology, Department of Gastroenterology, IRCCS Humanitas Research Hospital, Rozzano, 20089 Milan, Italy; 2Department of Biomedical Sciences, Humanitas University, Pieve Emanuele, 20090 Milan, Italy; 3Unit of Microbiology and Virology, IRCCS Humanitas Research Hospital, Rozzano, 20089 Milan, Italy; 4Department of Pharmaceutical Sciences, Università del Piemonte Orientale “Amedeo Avogadro”, Largo Guido Donegani, 28100 Novara, Italy; 5Wipro Life Sciences Lab, Wipro Limited, SJP2, Sarjapur Road, Bangalore 560035, Karnataka, India.; 6IBD Unit, Department of Gastroenterology, IRCCS Humanitas Research Hospital, Rozzano, 20089 Milan, Italy; 7Digestive Endoscopy Unit, Department of Gastroenterology, Humanitas Clinical and Research Center-IRCCS, Rozzano, 20089 Milan, Italy

**Keywords:** mucosal healing, polyphenols, biofilm, ulcerative colitis, Crohn’s disease

## Abstract

**Background:** Pomegranate (*Punica granatum*) can be used to prepare a bioactive extract exerting anti-inflammatory activities. Clinical studies demonstrated an improvement in clinical response in inflammatory bowel disease (IBD) patients when pomegranate extract (*PG*) was taken as a complement to standard medications. However, the molecular mechanisms underlying its beneficial effects are still scarcely investigated. This study investigates the effect of *PG* on bacterial biofilm formation and the promotion of mucosal wound healing. **Methods:** The acute colitis model was induced in C57BL/6N mice by 3% dextran sodium sulfate administration in drinking water for 5 days. During the recovery phase of colitis, mice received saline or *PG* (200 mg/kg body weight) by oral gavage for 11 days. Colitis was scored daily by evaluating body weight loss, bleeding, and stool consistency. In vivo intestinal permeability was evaluated by fluorescein isothiocyanate-conjugated dextran assay, bacterial translocation was assessed by fluorescence in situ hybridization on tissues, whereas epithelial and mucus integrity were monitored by immunostaining for JAM-A and MUC-2 markers. Bacterial biofilm formation was assessed using microfluidic devices for 24 or 48 h. Primary fibroblasts were isolated from healthy and inflamed areas of 8 IBD patients, and Caco-2 cells were stimulated with or without *PG* (5 μg/mL). Inflammatory mediators were measured at the mRNA and protein level by RT-PCR, WB, or Bio-plex multiplex immunoassay, respectively. **Results:** In vivo, *PG* boosted the recovery phase of colitis, promoting a complete restoration of the intestinal barrier with the regeneration of the mucus layer, as also demonstrated by the absence of bacterial spread into the mucosa and the enrichment of crypt-associated fibroblasts. Microfluidic experiments did not highlight a specific effect of *PG* on *Enterobacterales* biofilm formation, even though *Citrobacter freundii* biofilm was slightly impaired in the presence of *PG*. In vitro, inflamed fibroblasts responded to *PG* by downregulating the release of metalloproteinases, IL-6, and IL-8 and upregulating the levels of HGF. Caco-2 cells cultured in a medium supplemented with *PG* increased the expression of *SOX-9* and *CD44*, whereas in the presence of HGF or plated with a fibroblast-conditioned medium, they displayed a decrease in *SOX-9* and *CD44* expression and an increase in *AXIN2*, a negative regulator of Wnt signaling. **Conclusions:** These data provide new insight into the manifold effects of *PG* on promoting mucosal homeostasis in IBD by affecting pathogen biofilm formation and favoring the regeneration of the intestinal barrier through the regulation of the crosstalk between epithelial and stromal cells.

## 1. Introduction

Pomegranate (*Punica granatum* L.), belonging to the genus *Punica* L., Punicaceae family, is a plant known for its numerous bioactive compounds, present in the fruit pulp and peel, exerting several biological activities [1]. Polyphenols are one of the main classes of biologically active chemical substances in pomegranate and include flavonoids, condensed tannins, and hydrolyzable tannins [2]. Their poor absorption allows for high biodisponibility within the gastrointestinal lumen, where they serve as a substrate for bacterial fermentation. Polyphenols can be transformed by bacteria into bioactive beneficial metabolites, freely absorbed, and then recognized by several different cell types. Further biotransformation can also occur in the liver and the kidneys [3]. Besides exerting potentially prebiotic effects by selectively stimulating beneficial bacteria, *Punica granatum* L. polyphenols and their metabolites also display antioxidant, bactericide, anticarcinogenic, and anti-inflammatory effects [4,5,6]. All these properties have sparked scientific interest in this plant and stimulated the consumption of different pomegranate-related products, including whole fruit juice or extracts obtained using various parts of the plant (peel, seeds, and flowers) for the treatment and prevention of inflammatory conditions such as inflammatory bowel disease (IBD). Indeed, the consumption of 6 g of dry peel/day of *Punica granatum* peels for four weeks as complementary to standard medications improved clinical response in IBD patients [7].

IBD is a term including complex, chronic, relapsing remitting, not clearly defined, inflammatory diseases of the gut [8]. It is widely accepted that these conditions may be the result of a complex interplay between environmental factors, gut microflora, immunity, and genetic predisposition [9]. In IBD patients, loss of tolerance to the microbiota, in association with intestinal epithelial dysfunction, triggers an abnormal immune response which leads to progressive destructive damage and defective repair of the gastrointestinal tract. An effective and dynamic intestinal epithelial barrier is, therefore, crucial to conserve a compartmentalized microbe–host interaction and tissue homeostasis. In Crohn’s disease (CD) and ulcerative colitis (UC), the two most common forms of IBD, defects at different levels of intestinal barrier, from the mucus layer composition to the adhesion molecules to an altered microbial barrier (dysbiosis), have been described [10], indicating a leaky intestinal barrier as a primary dysfunction that leads and sustains chronic mucosal inflammation. Growing evidence supports polyphenols as potential modulators of gut barrier function through the expansion of certain bacteria that confer health benefits to the host and the downregulation of pro-inflammatory mediators. A higher intake of some polyphenols is associated with a lower risk of CD [11]. The mechanisms underlying these beneficial effects are currently under investigation, whereas no associations are reported between polyphenol intake and UC. Hydrolysis of punicalagin, an ellagitannin component, in ellagic acid (EA), which in turn is metabolized to urolithins by the intestinal microbiota, seems to play a crucial role in the *Punica granatum* extract properties. In vivo and in vitro studies, indeed, showed that urolithins exert anti-inflammatory effects modulating NF-kB and MAPKs signaling pathways by decreasing the secretion of pro-inflammatory mediators (IL-8, IL-18, COX-2, TNF-alpha, IL-1β, IL-6, and MCP-1) [12,13,14]. Moreover, ellagic acid and urolithins, respectively, strengthen the intestinal barrier and protect against inflammation-induced barrier dysfunction, reducing oxidative stress [15,16,17]. Thereby, advances in understanding the potential beneficial effects of dietary-derived polyphenols in preventing intestinal barrier dysfunction could be a strategy to control the disease. A proper repair of the epithelial barrier is supported by colonic subepithelial myofibroblasts, key players of mucosal wound healing that reside just beneath the epithelial compartment. By the modulation of extracellular matrix components and the release of growth factors, cytokines, and chemokines, myofibroblasts support the regeneration of the epithelial layer, favoring mucosal wound healing. Despite the growing number of studies demonstrating the beneficial properties of *Punica granatum* in IBD, there is no evidence of its effects during the regeneration of the mucosa. In this study, we assessed (i) the in vivo properties of *Punica granatum* extract to promote the healing of intestinal wounds, (ii) the effects on bacterial biofilm formation, and finally (iii) its effects on healthy and inflamed primary intestinal fibroblasts.

## 2. Materials and Methods

### 2.1. Human Tissue Collection

Primary intestinal fibroblasts were isolated from biopsy specimens of active IBDs (5 CD and 3 UC) and diagnosed based on clinical, biological, endoscopic, and histological criteria. A total of 62.5% were female, with a range of 27–68 years. Noninflamed and inflamed areas of mucosa were collected during an endoscopy procedure from all patients after they had given written informed consent. This study was approved by the ethical committee of the Humanitas Clinical and Research Center and conducted following national and international guidelines.

### 2.2. Isolation of Primary Intestinal Fibroblasts

Intestinal biopsies were collected in cool Roswell Park Memorial Institute (RPMI; Lonza, Basel, Switzerland) medium containing 10% FCS and 1% PSA and washed with Hank’s Balanced Salt Solution (HBSS; Gibco, Billings, MT, USA) containing 2% PSA. Digestion was performed with 750 µg/mL of Collagenase II (#LS004177, Worthington, Worthington, OH, USA) in RPMI medium complemented with 10% FCS at 37 °C for 20 min. The digested fragments were dissociated through 100-μm filters (BD falcon, Franklin Lakes, NJ, USA) with a 1 mL syringe plunger and washed with a 10 mL RPMI medium supplemented with 10% FCS and 1% PSA. The cell suspension obtained was filtered through 70-μm filters, then centrifuged to 300× *g* for 5 min discarding the supernatant. Cells were cultured in Gibco^TM^ M106 (#M106500, Billings, MT, USA) supplemented with 10% (*v*/*v*) Low Serum Growth Supplemented (#S00310, Gibco^TM^) and 1% (*w*/*v*) PSA at 37 °C, 5% CO_2_ and used between passages 2–8.

### 2.3. Cell Culture

Caco-2 (ATCC^®^ HTB-37^TM^) and HT-29CI.l6E cell lines were cultured in high glucose Dulbecco’s Modified Eagle Medium (DMEM, Lonza) supplemented with 10% of Fetal Calf Serum (FCS; Euroclone, Pero, Italy), 1 mM of sodium pyruvate (Euroclone), 2 mM of L-glutamine (Lonza), 0,1 mM of Non-Essential Amino Acids (NEAA; Lonza), and 10 µg/mL of penicillin/streptomycin with amphotericin (PSA; Lonza) at 37 °C in 5% CO_2_.

### 2.4. Cell Treatments

Caco-2 and HT29-CI.16E cell lines and human primary fibroblasts were treated for 48 h with *Punica granatum* (*PG*), 70.3% of total polyphenol content derived from Pomegranate Extract (PFE-045, Arjuna Natural Extracts Ltd., Aluva, India) suspended in sterile high glucose DMEM, with the following concentrations 5 µg/mL, 10 µg/m, 15 µg/mL, and 30 µg/mL. As a pro-inflammatory stimulus, Caco-2 and HT29-CI.16E cell lines and primary intestinal fibroblasts were stimulated for 48 h with 20 ng/mL and 20 *PG*/mL Tumor necrosis factor α (TNFα) (#210-TA, R&D system, Minneapolis, MN, USA), respectively. Caco-2 cells were stimulated with 200 ng/mL HGF (#294-HG-025, R&D system) for 2 h and in a 1:2 ratio for 24 h with conditioned IBD fibroblast medium (CFM), stimulated in the presence or absence of *PG*.

### 2.5. Animal Experiments

Female C57BL/6N mice aged 5–12 weeks old were purchased from the Charles River Laboratories. All animals were fed a standard diet and housed in specific pathogen-free conditions. All experiments were performed accordingly with the approval from the ethics committees of the Humanitas Clinical and Research Center, in agreement with national (D.Lgs 26/2014) and international animal care criteria; authorization 2034/2017-PR. An acute colitis model was induced in C57BL/6N mice by administration of 3% dextran sodium sulfate (DSS) (#0216011090, MP Biomedicals, Santa Ana, CA, USA) *ad libitum* in their drinking water for 5 days. During the recovery phase following the DSS challenge, mice were treated by oral gavage every 2 days with saline or *PG* 200 mg/kg body weight for 11 days. Mice were monitored during the entire experiment, and colitis was scored daily by the disease activity index (DAI), evaluating body weight loss, bleeding, and consistency of stools, according to criteria previously reported [10] and summarized in Table 1.

Endoscopic evaluation was performed before sacrifice to evaluate damages induced to the colonic mucosa during DSS treatment using the experimental endoscopy setup Coloview (Karl Storz, Tuttlingen, Germany). This procedure was performed under anesthesia. Colonic damage evaluation is based on several signs of inflammation: colon translucency and thickness, presence of fibrin, the granularity of the mucosal surface, morphology of the vascular pattern, and stool consistency.

### 2.6. Intestinal Permeability

To assess intestinal permeability, mice fasted for 4 h and were subjected to a single oral gavage with 60 mg/g fluorescein isothiocyanate-conjugated dextran (FITC Dextran 2 kDa; #52471, Sigma-Aldrich, St. Louis, MO, USA). Two hours after FITC-dextran administration, whole blood was collected and the fluorescent intensity was evaluated in the murine sera. FITC-dextran concentration was calculated using a FITC-dextran standard curve.

### 2.7. Microfluidic Assessment of Biofilm Formation

The following strains were used in this study: *Staphylococcus epidermidis* (ATCC 12228), *Escherichia coli* (ATCC 25922), *Klebsiella oxytoca* (NCTC 8167), and *Citrobacter freundii* (ATCC 8090). Bacteria were grown inside PDMS-glass microfluidic channel and exposed to different treatments. Microfluidic channels were fabricated using conventional soft lithography and rapid prototyping. Master molds were fabricated by patterning the negative photoresist SU-8 (MicroChem, Cape Town, South Africa) on silicon wafers. Positive replicas of the microfluidic channels were obtained by pouring polydimethylsiloxane (PDMS, Sylgard 184, Dow Corning, Midland, MI, USA) and agent curing 10:1 (*w*/*w*) on the master and degassed in a vacuum chamber to remove bubbles. The cured PDMS was peeled off, and connecting holes (inlets and outlets) were created using a biopsy puncher (1.5 mm). The PDMS channels were irreversibly bonded to a glass slide upon treatment with oxygen plasma. The devices were sterilized by UV irradiation before each experiment.

A single bacterial colony from pure culture on a Tryptone broth (TB) agar plate was transferred to a 15 mL sterile tube with 2 mL of TB and incubated at 37 °C in a shaker incubator. Bacterial suspensions that reached OD = 0.2 at 600 nm were injected directly into the microfluidic chamber. After 30 min of rest to allow bacterial adhesion to the surface, the device was connected to a flow system driven by a syringe pump, used to infuse fresh culture medium and treatments without introducing air into the chamber. All tests were performed at 37 °C using a live imaging incubator. A fully automated image acquisition routine recorded the position of bacteria on the surface of each channel every 4 min, for a total duration of the observation of at least 12 h.

### 2.8. Cytotoxicity Assay

Cell viability was assessed by a 3-(4,5-dimethylthiazol-2-yl)-2,5-diphenyltetrazolium bromide (MTT) assay. Caco-2, HT29CI.16E, and primary intestinal fibroblasts were plated at a density of 1.5 × 10^4^ cells/well into a 96-well flat plate, and 24 h after seeding, the cells were treated with 4 different concentrations of *PG* (5 µg/mL, 10 µg/mL, 15 µg/mL, and 30 µg/mL) or with 20% DMSO, each condition without (w/o) FBS. Some 48 h after the treatment, the cells were incubated for 4 h with 100 µL of MTT (500 µg/mL; #475989, Sigma-Aldrich) in DMEM w/o FBS. In order to solubilize the formazan crystal formed by MTT cell reduction, isopropanol with 0.04N HCL was added in a 1:1 volumetric ratio with MTT. Finally, a spectrophotometric reading was performed at 490 nm to quantify the formazan dye product. The absorbance values were normalized to the values of absorbance of the medium alone and reported as a percentage of mitochondrial activity.

### 2.9. Trans-Epithelial Electric Resistance

Caco-2 cells were seeded at a density of 10^4^ cells/well into a collagen-coated (50 µg/mL; Corning, Corning, NY, USA) upper chamber of a 6-well transwell filter (pore diameter, 0.4 µm; Costar). The transepithelial electric resistance (TEER) was measured with a voltmeter (Millicels; Millipore, Bedford, MA, USA) before the stimulation (T = 0) and daily during 60 h after stimulation with LPS (10 µg/mL; #L4005, Sigma-Aldrich), *PG* (60 µg/mL) and the combination of the two stimuli (LPS + *PG)*. The TEER value was calculated by subtracting the blank value. Finally, the percentage of change in TEER was evaluated with time 0 for each condition.

### 2.10. Real-Time PCR

Total RNA extraction from Caco-2 and primary intestinal fibroblasts was performed using the Rneasy Mini kit (#74104, Qiagen, Hilden, Germany) according to the manufacturer’s instructions. cDNA synthesis was carried out using 200 or 1000 ng of total RNA with the High Capacity cDNA Reverse Transcription Kit (Applied Biosystems, Waltham, MA, USA). Quantitative real-time PCR was performed for 10 ng of cDNA using SYBR Green Master Mix (Applied Biosystems) and detected with Viia7 Detection system (Applied Biosystems). Glyceraldehyde-3-phosphate dehydrogenase (*GAPDH*) gene has been used as a housekeeping gene, and the relative mRNA level was expressed as 2^−Δ^*^C^^t^*. The specific primer sequencing used is reported in Table 2.

### 2.11. Collagen Contraction Assay

Contraction capacity was assessed by contraction assay on primary intestinal fibroblasts derived from IBD patients. Briefly, 400 µL of 1.5 × 10^5^ cell suspension in basal DMEM was mixed with 200 µL of Collagen type I from rat tail (1.1 mg/mL; #354236, Corning) in a 1.5 mL Eppendorf tube. Then, 5 or 6 µL of NaOH (1N, Merck, New York, NY, USA) was added to equilibrate the pH, and 500 µL of the mixture was transferred into a 24-well plate. After 20 min at room temperature, when the gel solidification occurred, the cells were treated for 48 h with *PG* (5 µg/mL) or tumor growth factor-β (recombinant human TGF-β1, #100-21, Peprotech, Cranbury, NJ, USA), as a contraction−inductor factor, performing two stimulations of 5 ng/mL per day. The gel contraction was quantified by measuring the minor and major diameters and calculating the area of the ellipse.

### 2.12. Enzyme-Linked Immunosorbent Assay (ELISA)

Culture supernatants derived from Caco-2 and primary intestinal fibroblasts, treated for 48 h with *PG* (5 µg/mL) or TNFα (20 ng/mL and 20 *PG*/mL respectively) or the combination of the two stimuli, were analyzed for Interleukine-8 (IL-8), IL-6, and IL-33 content in duplicate exploiting commercially available ELISA kits (R&D Systems, Minneapolis, MN, USA), following the manufacturer’s protocol.

### 2.13. Bio-Plex Multiplex Immunoassay

The presence in murine serum of IL-4, IL-6, IL-10, IL-13, IL-17, TGF-β, TNF-α, Keratinocyte-derived chemokine (KC), IFN-γ, IL-5, Monocyte chemoattractant protein-1 (MCP-1), IL-12p40, and IL-12p70 were quantified by Bio-Plex Pro^TM^ Mouse cytokine Assay following the manufacturer’s instructions (Biorad, Hercules, CA, USA). Serum was diluted 1:3 prior to analysis. Bio-Plex 200 system was employed to read and analyze the concentration of the cytokines.

### 2.14. Histological Preparation and Sirius Red Staining

At the end of the experiment, the murine colon was explanted, measured in length, and embedded in an OCT compound. For histological assessment, 2 µm-thick sections were stained with hematoxylin and eosin. Collagen deposition was evaluated by Sirius Red staining performed on 6 μm-thick frozen tissue sections. A second-harmonic generation analysis of collagen fibers was conducted by imaging Sirius red-stained slides with the Olympus BX-51 fluorescent microscope using the U-ANT and U-POT polarizers (Olympus, Tokyo, Japan), and images were quantified with ImageJ27.

### 2.15. Immunofluorescence Staining on Frozen Tissue Sections

Frozen murine colon sections collected on SuperFrost glass slides (Thermo Scientific, Waltham, MA, USA) were fixed in 4% (*v*/*v*) paraformaldehyde for 10 min at room temperature and permeabilized with 0.1% (*v*/*v*) Triton X-100 in PBS (all from Sigma-Aldrich) for 20 min at room temperature. The sections were blocked for 1 h with 1% (*w*/*v*) BSA, and incubated with the following primary antibodies for 1 h at room temperature: rabbit anti-mouse MUC2 (1:25; #sc-15334, Santa Cruz, Santa Cruz, CA, USA), mouse anti-mouse α-SMA (1:200; #a2547, Sigma-Aldrich), goat anti-mouse JAM-A (1:20, #AF1077, R&D) followed by 40 min of incubation at room temperature with Alexa Fluor 488-conjugated goat anti-rabbit (1:500; #A11034, Invitrogen, Waltham, MA, USA), Alexa Fluor 594-conjugated goat anti-mouse (1:500; #A11005, Invitrogen) and Alexa Fluor 488-conjugated donkey anti-goat (1:500; #A11055, Invitrogen), respectively. DAPI (1:25,000, Invitrogen) was used for nuclear staining. Coverslips were finally mounted with Fluorescence Mounting Medium (#S3023, Dako, Santa Clara, CA, USA). The images were acquired by Leica SP8 Confocal Microscopy using a 60X objective and analyzed using ImageJ.

### 2.16. Fluorescence In Situ Hybridization on Frozen Tissues

The frozen section (10 µm) of the colon was fixed in a Metacarnoy fixation solution (3 parts of MeOH and 1 part of Acetic acid) at room temperature for 4 h, washed with 70% EtOH for 5–10 min to remove the cryoblock, and then dried. An overnight incubation at 50 °C of 62.5 ng/µL of diluted FISH probes was performed for FISH staining. Probes were diluted in a hybridization buffer containing 0.9 M NaCl, 0.2 M of Tris HCL pH = 7.4, 0.1% of SDS 20%, and H_2_O. The slides were washed with wash buffer (20 mM TrisHCL + 0.9 NaCl) at 50 °C for 20 min and blocked with 2% BSA in wash buffer at room temperature for 20 min. DAPI (1:25,000, Invitrogen) was used for nuclear staining. The section was mounted with Prolong gold and analyzed with a laser scanning confocal microscope (Fluoview FV1000; Olympus, Tokyo, Japan). Images were acquired with an oil immersion objective (100 × 1.4 NA Plan Apochromat; Olympus).

### 2.17. Immunofluorescence Staining on HT29CI.16E Cell Line

To allow cell polarization, HT29CI.16E cells were seeded with a density of 5 × 10^4^ cells on a Transwell membrane ThinCert^TM^ (0.4 µm; #662641, Greiner Bio-One, Cassina de Pecchi, Italy) in a 24-well plate. When cells reached 80% confluence, they were treated for 48 h with *PG* (5 µg/mL) or TNFα (20 ng/mL) or a combination of the two stimuli. After 48 h, HT29CI.16E were fixed in 4% (*v*/*v*) paraformaldehyde for 10 min at room temperature, and the transwell membrane with the cells on top was cut and placed on SuperFrost glass slides. Then, cells were permeabilized with 0.1% (*v*/*v*) Triton X-100 in PBS (all from Sigma-Aldrich) for 20 min at room temperature and incubated for 1 h with 1% (*w*/*v*) BSA and incubated with mouse anti-human MUC2 conjugated with Alexa Fluor 488 (1:100; #sc-515032 AF488, Santa Cruz) overnight at 4 °C, followed by DAPI (1:20,000, Invitrogen) counterstaining. Coverslips were finally mounted with Fluorescence Mounting Medium (#S3023, Dako). The images were acquired by Leica SP8 Confocal Microscopy using a 60X objective and analyzed by using ImageJ.

### 2.18. Western Blotting

Protein extraction from cell pellets was performed using RIPA buffer [10 mM Tris-Cl (pH 8.0), 1 mM EDTA, 140 mM NaCl, and 1% Triton X-100, 0.1% (*v*/*v*) sodium deoxycholate, 0.1% SDS] supplemented with protein inhibitor cocktail (1:50; #A32963, Thermo Scientific) and the phosphatase inhibitors, 1 mM Sodium orthovanadate. Proteins extracted were quantified by DC Protein Assay (Biorad) following the manufacturer’s protocol. 20 and 30 µg of protein extracts (fibroblasts and cell lines, respectively) were resolved by 4–20% stain-free precast SDS-PAGE (Bio-Rad). After incubation with 5% (*w*/*v*) non-fat milk or 5% bovine serum albumin (BSA; Sigma-Aldrich) in Tris Buffered Saline with Tween 20 (TBS-T; 10 mM Tris, pH 8.0, 150 mM NaCl, 0.5% (*v*/*v*) Tween-20) for 1 h, the membranes were incubated with antibodies against rabbit anti-MUC2 (1:1000; #PA5103083, Invitrogen), rabbit anti-JAM-A (1:250; #36-1700, Invitrogen), rabbit anti-p44/42 MAPK (ERK1/2) (1:1000; #4695, Cell Signaling Danvers, MA, USA), rabbit anti-Phospho-p44/42 MAPK (ERK1/2) (Thr202/Tyr204) (1:1000; 4370, #Cell Signaling), mouse anti-AKT (1:1000; #2920, Cell Signaling), rabbit anti-Phospho-AKT (Ser473) (1:1000; #4060, Cell Signaling), rabbit anti-STAT3 (1:1000; #12640, Cell Signaling), rabbit anti-Phospho-STAT3 (Tyr705) (1:1000; #9145, Cell Signaling), mouse anti-GAPDH(1:5000; #AMAB91153, Sigma-Aldrich) overnight at 4 °C. Then, after three washes with TBS-T, the membranes were incubated with horseradish peroxidase-conjugated anti-rabbit (1:1000; #HAF008, R&D system) or goat anti-mouse (1:2000; #sc-2005, Santa Cruz Biotech, Dallas, TX, USA) antibodies for 1 h at room temperature, followed by development using Immobilon ECL Ultra Western HRP Substrate (#WBULS0500, Millipore, Burlington, MA, USA) or Super Signal^TM^ West Dura extended duration Substrate (#34096, Thermofisher, Waltham, MA, USA) according to the manufacturer’s protocols. Finally, the immunoreactivity was detected by Chemidoc (Bio-Rad Laboratories) and quantified by Image J Fiji 1.53t software. Phosphorylated AKT, STAT3, and ERK were normalized on total AKT, STAT3, and ERK, respectively.

Proteins in the cell culture supernatant were concentrated using StrataClean Resin (1:30, #400714, Agilent, Santa Clara, CA, USA). Proteins were separated on 4–20% stain-free precast SDS-PAGE and transferred onto nitrocellulose membrane according to the manufacturer’s protocols (Bio-Rad). After incubation for 1 h at room temperature with 5% (*w*/*v*) non-fat milk in TBS-T, the membranes were incubated with the antibody against rabbit anti-human HGFα (1:200; #sc7949, Santa Cruz Biotech.) overnight at 4 °C. Then, membranes were incubated with horseradish peroxidase-conjugated anti-rabbit (1:1000, Santa Cruz) for 1 h at room temperature. The development and the detection of the immunoreactivity were performed as reported above. HGF was normalized on total proteins.

### 2.19. Statistical Analysis

All the results were analyzed using GraphPad Prism 6 software and expressed as mean ± SEM. One-way or two-way ANOVA were used for the comparison of multiple groups, whereas paired and unpaired *t*-test were for the comparison between two groups. Statistical significance was defined as *p* < 0.05.

## 3. Results

### 3.1. Pomegranate Extract Accelerates the Recovery Phase of Acute Colitis

Several in vivo studies reported the beneficial effects of *PG* in dampening intestinal inflammation in the acute phase of DSS-induced colitis [13,18]. However, no studies focused on the recovery phase, which is crucial for the proper repair of the mucosal epithelial barrier and re-establishment of intestinal homeostasis. To address this aspect, moderate acute colitis was first induced by ad libitum administration of 3% DSS for 5 days. DSS treatment was successful in promoting an inflammatory state, as evidenced by a 10% decrease of initial body weight (Figure 1A) and an increase in the bleeding score (Figure 1B). Starting from day 6, DSS treatment was replaced with regular water, and mice were treated with 200 mg/kg of the whole extract of *Punica granatum* (*PG*) or saline as the control group. Mice receiving *PG* showed rapid recovery of the initial body weight (Figure 1A) in association with a reduction of bleeding score compared to control mice (Figure 1B). *PG*-treated mice also displayed a longer colon length than the saline-treated group (Figure 1C). Endoscopic monitoring of the intestinal wall revealed a complete recovery of mucosal damage by day 16 in the *PG* group. Indeed, the mucosa appeared transparent with a clear organization of vascular pattern, without traces of fibrin and alterations of the mucosal surface. On the contrary, the saline-treated group displayed a moderate grade of wall thickening with bleeding areas (Figure 1D).

On the day of the sacrifice (day 16), serum was collected from all mice, and the expression of multiple inflammatory mediators was analyzed using a multiplex bead-based assay. As expected, *PG* reduced, albeit not significantly, the levels of KC, a chemo-attractant responsible for recruiting neutrophils, and monocyte chemo-attractant protein-1 (MCP-1), a potent chemotactic factor for monocytes (Figure 1E). Interestingly, mice treated with *PG* showed significantly increased levels of IL-5, IL-10, and IFN-γ with respect to healthy controls and saline-treated mice, while no differences were observed in the levels of TNF-α, IL-12p40, IL-12p70, IL-13, IL-6, between the groups (Figure 1E). Altogether, these results corroborate the beneficial properties of *PG* in promoting recovery from mucosal damage.

### 3.2. PG Improves Restoration of Intestinal Barrier In Vivo

In order to gain insight into the mechanisms by which *PG* improves mucosal healing, intestinal permeability was quantified as a measure of gut barrier function by performing a fluorescently labeled small molecule (FITC-dextran) assay, which is an indirect measure of total intestinal permeability. On day 16 after DSS treatment, mice were given an oral gavage of FITC-dextran (44 mg/mL), and fluorescence was quantified in serum in all mice after 4 h. Mice treated with *PG* and healthy control mice displayed low levels of FITC-dextran, while in the saline-treated group, the presence of dye was, despite not being significant, higher, indicating an incomplete restoration of intestinal epithelial integrity (Figure 2A). In line with this finding, the staining for Junctional Adhesion Molecule A (JAM-A), a tight junction protein implied in the assembly and barrier function, was strong and uniform in the epithelial layer and crypts of the *PG* group. In saline-treated mice, staining was weaker, showing a spotty positivity of JAM-A along the epithelial layer (Figure 2B). As the dysfunction of the intestinal epithelial barrier promotes the translocation of bacteria and bacterial-related products into the mucosa, activating an immune response, fluorescence in situ hybridization (FISH) was performed to visualize bacterial rRNA in the colonic mucosa. As expected, after five days of DSS treatment, the mucus layer was lost, and bacteria were present within the mucosa immediately below the epithelial layer, where host-microbe interactions begin when an impaired barrier function occurs (Figure 2C). Ten days of *PG* treatment promoted a complete restoration of epithelial integrity with the formation of a mucus layer. Indeed, mice treated with *PG* showed strong staining for mucin-2 (MUC-2), a major gel-forming mucin of the colon forming a protective gel barrier, which prevents the colonic epithelium of the host from coming into direct contact with the microbiota, and no traces of bacterial spread into the mucosa. On the contrary, the saline-treated group displayed an absence of mucus layer with a weak expression of MUC-2 within the crypts, even though the FISH assay was negative (Figure 2C,D), confirming an incomplete restoration of the epithelial barrier in this group.

### 3.3. PG Enhances Epithelial Barrier Integrity In Vitro

To understand whether *PG* directly promoted the production of MUC2 in epithelial cells, in vitro assays were conducted on the human mucin secreting colonic epithelial cell line (HT29-C1.16E). After polarization over two weeks on transwell supports (Figure 3A), the cells were first tested for *PG* toxicity using a tetrazolium reduction assay that measures enzymatic activity as a marker of cell viability. After 48 h of culturing, the cells resulted viable for all tested concentrations (5–30 μg/mL) of *PG* (Figure 3B). Secondly, MUC-2 expression was quantified after 48 h by WB and immunofluorescence assays at the lowest concentration (5 μg/mL). Both assays demonstrated the capacity of *PG* to induce MUC-2 production (Figure 3C,D). According to a previous study [19], we confirmed the anti-inflammatory properties of *PG*. Indeed, in Caco-2 cells, *PG* reverted the increased levels of IL-8 induced by TNF-α stimulation (Figure 3E). The effects of *PG* on JAM-A expression were then assessed using monolayers of Caco-2 cells. After 48 h of culture, the group cultured with *PG* (5 μg/mL) showed enhanced JAM-A expression, restoring TNF-α mediated downregulation (Figure 3F). Since Transepithelial electrical resistance (TEER) reflects the integrity of tight junction-mediated paracellular permeability, TEER measurements were performed on the Caco-2 monolayer without cell damage or after 24 h of LPS stimulation. TEER values significantly increased from 100 ± 7 to 300 Ω/cm^2^ in the presence of *PG* with and without LPS (Figure 3G). Overall, these results corroborate the beneficial properties of *PG* in regulating the gut barrier by modulating the mucus layer and junctional adhesion protein production.

### 3.4. PG Modulates Gut Microbial Biofilm

The loss of the mucus layer and of the integrity of the epithelial barrier promotes the access of bacteria to the colonic epithelium, which in turn alters physiological microbial community relationships, composition, and activity. The formation of complex biofilms directly attached to the colonic epithelium promotes alterations in its integrity, ultimately leading to intestinal inflammation. We, therefore, tested the effects of pomegranate-derived polyphenols on biofilm formation by three species of *Enterobacterales*, including a commensal isolate of *Escherichia coli* and isolates of *Klebsiella oxytoca*, and *Citrobacter freundii*. As a comparison, we also tested biofilm formation capacity in the presence of *PG* of a commensal Gram-positive bacterium, the *Staphylococcus epidermidis*. Interestingly, the addition of *PG* extract significantly reduced biofilm formation ability by the gram-positive *S. epidermidis* at 48 h vs. non-supplemented medium (untreated) (Figure 4A). A trend of reduction was also observed for *C. freundii* even though not statistically significant (Figure 4D), while no no differences were observed in other *Enterobacterales* biofilms (Figure 4B,C).

### 3.5. PG Promotes Fibroblast Accumulation during the Recovery Phase of Acute Colitis

The recovery of barrier integrity involves various nonepithelial cells, such as myofibroblasts, which are known to be key players in the repair mechanisms that involve inflammatory processes. Importantly, myofibroblasts influence epithelial cell migration and proliferation through degradation and formation of extracellular matrix components and thus modulate intestinal epithelial healing. To investigate whether *PG* improved the recovery from the colitic phase influencing myofibroblast activity, a colonic section from mice treated with *PG* or saline was stained for alpha-Smooth Muscle Actin (α-SMA), a marker of fibroblast activation. After 10 days of *PG* treatment, the staining for α-SMA was strongly positive within the intestinal lamina propria and in the areas surrounding the crypt epithelium (Figure 5A lower panel), whereas in the saline-treated group, it appeared weakly positive (Figure 5A upper panel). To exclude that *PG* caused fibrogenic processes by promoting activation of fibroblasts and excessive collagen deposition, essential for proper wound healing, the presence of collagen was evaluated by Sirius Red staining and then by polarized light microscopy that provides complementary information about forms of collagen. Both approaches revealed that *PG*, despite promoting expansion of fibroblasts, does not induce an accumulation of collagen within the intestinal mucosa or in tunica mucosa and submucosa (Figure 5B lower panel). Indeed, Sirius Red staining provided similar results between *PG* and saline-treated mice, and the birefringence pattern of collagen fibers did not change between the two groups (Figure 5B), thus highlighting that *PG* does not promote the accumulation and maturation of collagen fibers typical of fibrotic processes.

### 3.6. PG Reverts Fibroblast Activation

To investigate further the effects of *PG* on the activation of fibroblasts, primary intestinal fibroblasts isolated from noninflamed and inflamed areas of IBD patients (UC, *n*= 3 and CD, *n*= 5) were cultured in the presence and/or not of *PG* at a concentration of 5 µg/mL, identified through an MTT assay as the optimal concentration to maintain a cell viability of 100% over 48 h (Figure 6A). The release of IL-8 and IL-6 cytokines was tested, as both cytokines are known to promote fibroblast activation and proliferation. No difference was observed between healthy and IBD groups before and after *PG* stimulation in terms of cytokine release whereas *PG* significantly reverted TNF-α dependent release of IL-6 and IL-8 in all groups of fibroblasts (Figure 6B). Indeed, a paired-sample analysis performed after 48 h of stimulation with TNF-α and *PG* clearly showed the modulatory effects of *PG* in all samples.

Accordingly, *PG* did not per se promote the expression of proteolytic enzymes, in particular of metalloproteinases (MPPs) 1 and 3, key players of activated fibroblasts , but reverted the synthesis of MPPs 1 and 3 induced by TNF-α (Figure 6C). These effects may in part be mediated by the synthesis of TIMP1, an inhibitory molecule of MPPs, that, despite not significantly *PG* in healthy fibroblasts, was able to revert TNF-α modulation (Figure 6C).

During tissue repair, aberrant activation of fibroblasts allows their differentiation into a more contractile phenotype known as myofibroblasts, which deposit excessive collagen in the interstitium, thus contributing to fibrotic processes. To ascertain whether *PG* contributed to this process, a collagen gel contraction assay was performed with the aim of measuring fibroblast contractility. Thereby, all groups of fibroblasts were stimulated for 48 h with or without transforming growth factor beta (TGF-β, a profibrotic factor able to promote fibroblast contractility) at the concentration of 5 ng/mL per day and in the presence or in the absence of *PG* (5 μg/mL) (Figure 6D).

In all IBD samples, the stimulation with *PG* reverted significantly the contractile forces of fibroblasts generated by TGF-β. Taken together, these results demonstrate that *PG* per se does not activate intestinal fibroblasts, but importantly it downregulates fibroblast activation, extracellular matrix production and contraction.

### 3.7. PG Controls Intestinal Epithelium Repair by Regulating Fibroblast Activity

Intestinal fibroblasts play a key role in the mucosal healing processes by promoting epithelial renewal and cell proliferation through the secretion of growth factors. To assess whether this function could be influenced by *PG*, Caco-2 cells were cultured for 24 h with a conditioned medium (CFM) derived from IBD fibroblasts stimulated in the presence or absence of *PG* (Figure 7A). The expression of *SOX-9* and *CD44* was then evaluated, as these markers are highly expressed in the epithelial stem cell zone, which modulates stem/progenitor cell proliferation and differentiation [20,21]. Considering that polyphenols exert health-protective effects on epithelial cells by improving the epithelial barrier and decreasing anti-inflammatory mediators such as IL-8 (Figure 3E), we analyzed first the effects of *PG* directly on Caco-2 cells and then in the presence of CFM. Interestingly, Caco-2 cells stimulated with *PG* showed a significant increase in both *SOX-9* and *CD44* expression compared to untreated Caco-2 cells (Figure 7B,C). Conversely, *SOX-9* and *CD44* levels significantly dropped off in the presence of the CFM derived from fibroblasts stimulated with *PG*. Importantly, *SOX9* and *CD44* expression also diminished—although not significantly—with conditioned medium derived from untreated fibroblasts compared to untreated Caco-2 suggesting that fibroblasts could exert a negative feedback loop that counteracts epithelial regenerative processes. In line with this, *AXIN2*, a negative regulator of the Wnt/β-catenin signaling pathway, which prevents proliferative responses, was strongly downregulated by *PG* stimulation on Caco-2 cells, whereas CFM derived from IBD fibroblasts stimulated with *PG* reverted *AXIN2* expression to baseline levels (Figure 7D).

These results support the important role of *PG* in the repair of the epithelial barrier and unveil its function in arresting epithelial regeneration through fibroblast-derived paracrine factors. Since IL-33 and Hepatocyte Growth Factor (HGF) have been demonstrated as factors released by fibroblasts involved in mucosal healing processes [22,23], the effect of *PG* on the release of both factors was tested by quantifying the levels of IL-33 and HGF in the supernatants of IBD fibroblasts stimulated for 48 h in presence or the absence of *PG* (5 μg/mL). IL-33 resulted undetectable in all experimental groups , whereas HGF levels significantly increased after *PG* stimulation (Figure 7E). To verify the involvement of HGF in counteracting the epithelial regeneration, Caco-2 cells were stimulated with recombinant human HGF (200 ng/mL) for 2 h, and *SOX-9*, *CD44,* and *AXIN2* mRNA expression were then analyzed. In agreement with CMF effects, HGF negatively modulated *SOX-9* and *CD44* compared to *PG* and upregulated *AXIN2*, albeit with lower power (Figure 7B–D). These findings support the key role of *PG* in promoting epithelial proliferation and in regulating the crosstalk between epithelial cells and fibroblasts, providing a negative feedback loop preventing excessive proliferation.

To assess which signaling pathways activated by *PG* orchestrate the different effects on Caco-2 and fibroblasts, we quantified the phosphorylation of STAT3, ERK, and AKT pathways, previously demonstrated to mediate the beneficial effects of polyphenols. While the phosphorylation of AKT and STAT3 signaling was upregulated in Caco-2 cells after *PG* (Figure 7F,G), the same signaling pathways were significantly inhibited in fibroblasts (Figure 7H,I) after 48 h of *PG* stimulation compared to untreated fibroblasts from IBD patients. No changes were induced by *PG* in the phosphorylation of ERK in both Caco-2 cells and fibroblasts (Figure 7F–I). These results indicated that epithelial cells and fibroblasts respond to *PG* through common signaling pathways but in opposite ways.

## 4. Discussion

IBD is a chronic, relapsing, and remitting condition for which no cure is available. Currently, the increasing plethora of pharmacological treatments has as an ultimate goal to reduce inflammation, favor mucosal healing, cease symptoms, and reduce flare ups. In the last few decades, there has been a growing interest in natural compounds such as polyphenols for the development of nutraceutical and pharmaceutical products as dietary supplements due to their anti-inflammatory, antidiabetic, and anticancer properties. These beneficial properties have encouraged the consumption of foods rich in polyphenols, including tea, herbs, spices, dark chocolate, red wine, nuts, and fruits such as berries, pomegranates, and grapes. Interestingly, a higher intake of some polyphenols has been associated with a lower risk of CD, but there are no associations with polyphenol intake for UC [24]. The predominant anti-inflammatory mechanism of polyphenols in IBD seems attributed to the inhibition of TLR4/NF-κB-mediated signaling pathways and the downregulation of expression of pro-inflammatory mediators [25]. However, most of the studies have been conducted on the acute inflammatory phase, and no attention was paid on the effects of polyphenols during the resolution phase. Here, we demonstrated for the first time that polyphenols not only dampen inflammation but accelerate the recovery of mucosal damage by acting on immune and non immune cells. Several polyphenol-rich extracts, like those obtained from *Punica granatum*, *Boswellia serrata*, or *Curcuma longa* are known to have in vitro anti-inflammatory and beneficial effects on intestinal epithelial cells, strengthening intestinal barrier and protecting against inflammation-induced barrier dysfunction [6,26,27]. Accordingly, our data showed a strong capacity of *PG* in affecting biofilm formation by pathogens, promoting the expansion of commensals, and accelerating the recovery of barrier function impaired by DSS-induced colitis. Colitic mice, after DSS removal and 10 days of daily administration of the *PG* displayed a faster recovery of mucosal damage supported by the amelioration of clinical, endoscopic, and microscopic parameters. In fact, the treatment improved wound re-epithelialization of mucosa, as evidenced by the absence of mucosal bleeding when compared to the saline-treated group. Furthermore, *PG* increased the levels of IL-5, IL-10, and INFγ and dampened pro-inflammatory mediators such as KC, MCP1, IL12p40, and IL12p70, whereas the secretion of TNF-α and IL-13 remained high in both *PG* and saline-treated groups compared to healthy mice. IL-10, secreted by T helper cells, macrophages, dendritic cells and neutrophils is highly relevant to IBD, as also demonstrated in IL-10 −/− deficient mice that spontaneously develop very early colitis. IL-10 exerts a protective function by regulating the different phases of the inflammatory process and promoting intestinal epithelial cell proliferation and wound repair [28]. The capacity of *PG* to induce IL-10 release may justify the wound re-epithelialization of mucosa and the restoration of the mucus barrier, increasing mucin (MUC)-2 production observed in the *PG*-treated mice. IL-10, indeed, has been identified as a factor that promotes MUC-2 release, and its loss leads to a defective colonic mucin synthesis [29]. MUC-2 secreted by intestinal goblet cells is essential for regulating gut microbiota homeostasis and preventing bacterial translocation. An impaired MUC-2 secretion is associated with IBD [30] and microbial dysbiotic status [31]. Nevertheless, our in vitro experiments on human goblet cell-like (HT29C1.16E) clearly indicated that *PG* enhances MUC2 production also by IL-10 in an independent way. It is plausible that this effect could result from the induction of multiple pathways, including epigenetic regulation [32]. Although further studies are needed, *PG* likely activates multiple mechanisms regulating the secretory function of intestinal goblet cells and mucin levels. In addition to the increase in MUC-2, *PG* improved epithelial integrity by enhancing the expression of JAM-A, a tight junction crucial in maintaining intestinal barrier integrity [10] thus restricting the entry of microorganisms and toxins. Therefore, *PG* stimulates gut repair processes by stabilizing intestinal barrier integrity, affecting mucus layer and paracellular permeability.

An intact and functional epithelial barrier is important for maintaining the balance of microbial species, where normal commensal microbiota attach to the surfaces of the epithelium and translocation of pathogens is prevented. The loss of commensals allows the establishment and colonization of pathogens by taking advantage of the surfeit of nutrients [33] and by forming a biofilm, a self-produced matrix of extracellular polymeric substances (EPS). Microbial EPS synthesized by bacteria consists of polysaccharides, proteins, glycoproteins, glycolipids, and, in some cases, extracellular DNA, which determine the biofilm properties such as penetration of antimicrobial agents and the immune response of the host. The factors that influence gut colonization and the consequences of improper factors leading to dysbiosis and biofilm formation in IBD patients are not fully known.

This work shows that 48 h exposure to *PG* significantly inhibits the biofilm formation by *Staphylococcus epidermidis*, a common human skin commensal and opportunistic pathogen, and, to a lesser extent, also of *Citrobacter freundii*, a member of the Enterobacterales genus. As biofilm is considered the most relevant virulence determinant of coagulase-negative staphylococci (as *S. epidermidis*) when causing opportunistic infections [34], these results may prompt further studies on the role of *PG* on gram-positive biofilm. The lack of significant effect on *E. coli* and *K. oxytoca* biofilms can be related to a number of elements, including a different cellular structure (i.e., gram positive vs. gram negative) or biofilm composition, but also to the exposure time. As biofilm formation requires several steps to be established [35,36], we cannot exclude that 48 h observation is too short a timeframe to observe a significant effect. Another possible factor to take into account is *PG* dosing. Plant polyphenols represent a first line of defensive barrier to the colonization and invasion of pathogens by affecting bacterial regulatory mechanisms such as quorum sensing or other global regulator systems [37,38,39]. The antibiofilm capacity may be due to the content of ellagic acid, which alters the expression of genes linked to biofilm production [40]. Previous studies showed an inhibitory effect of ellagic acid on biofilm synthesized by other microbial species, including *E. coli*, but at 15 to 40 µg/mL concentrations, therefore at higher concentrations than those used in this study. Considering the faster recovery of the intestinal epithelial barrier and the enhancement of the fibroblast population in the proximity of the crypts in mice treated with *PG*, this investigation was mainly focused on the regenerative effects of *PG* in promoting mucosal repair and its effects on fibroblasts. Therefore, the concentration of 5 µg/mL was explored in vitro as it turned out to be nontoxic for the cells. Interestingly, at this concentration, primary fibroblasts isolated from inflamed and noninflamed mucosa of IBD patients responded to *PG* reverting TNF-α dependent release of IL-6 and IL-8, and reducing metalloproteinase expression, indicating the capability of *PG* capability to revert fibroblast activation. Fibroblasts are professional contractile cells that transiently remodel the connective tissue by contracting and pulling on the extracellular matrix, promoting the repair of the epithelium following damage. Within 10 days after injury, activated fibroblasts allow the regeneration of new tissue. Upon completion of this process, activated fibroblasts die. Nevertheless, if they persist longer, the excessive contraction and the resulting overremodeling of the ECM organization with increasing collagen deposition lead to the fibrotic process [41]. *PG* prevented in vitro fibroblast-mediated collagen gel contraction, an assay usually performed to test the contractile capacity of fibroblasts. The inhibitory effects of *PG* on TGF-β dependent contraction excluded an excessive repair process fueled by polyphenols. Accordingly, in vivo, despite promoting the expansion of fibroblasts, *PG* did not induce an accumulation of collagen or a difference in formed collagen fibers within the intestinal mucosa or in tunica mucosa and submucosa. These data confirm that *PG* does not promote the accumulation and maturation in the gut of collagen fibers typical of fibrotic processes. Consistently with this, HGF, which exerts antifibrotic effects by modulating TGF-β signaling [23], was upregulated in primary fibroblasts stimulated with *PG*. HGF is a paracrine soluble factor that supports intestinal tissue repair controlling epithelial cell proliferation. Resident fibroblasts, indeed, besides producing extracellular matrix and participating in tissue ECM maintenance in homeostasis and remodeling after injury, play an important role in the regeneration of epithelial cells by secreting cytokines and several growth factors [41], including IL33 and HGF [22,23]. Different fibroblast subsets have been identified in the gut; those located around the crypts drive epithelial stem cell niche maintenance and crypt proliferation through Wnt signaling [42]. Although further studies are needed to explore the mechanisms more deeply, the data presented in this work raise the hypothesis that fibroblasts respond to *PG* by secreting paracrine factors that negatively control epithelial regeneration. *PG* boosts the expression of *CD44* and *SOX-9* genes on Caco-2 cells, both Wnt target genes, highly expressed in intestinal crypts by stem cells and immature cells constituting the proliferative compartment of the epithelium in the regeneration [43]. Accordingly, *AXIN-2*, a negative regulator of Wnt that silences the signaling pathway, decreased. By contrast, when Caco-2 cells were cultured with fibroblast-conditioned medium after pretreatment with *PG*, *CD44*, and *AXIN*-2 expressions were completely reverted, whereas *SOX-9*, which is not only a target but also a strong inhibitor of the Wnt/β-catenin signaling pathway [44] thus counteracting an excessive epithelial proliferation, gradually returned towards baseline levels. HGF was able to mimic the effects mediated by CFM, but not at the same entity, indicating that other factors could be involved by activating negative feedback on epithelial cells. Interestingly, the different effects exerted by *PG* on epithelial cells and fibroblasts could be explained by the differential activation of AKT and STAT3 signaling transducers between the two cell populations. Both AKT and STAT3 signaling were activated on epithelial cells and inhibited on fibroblasts by *PG*. AKT is a signal transduction pathway that promotes survival and cell proliferation via the activation of several transcription factors, including Sox-9 [45]. Emerging evidence supports the role of CD44 as a driver of AKT activation [46]. It is plausible that the epithelium responds to *PG* by increasing the expression of *CD44*, which in turn promotes a cascade activation of AKT-dependent target genes, including *SOX-9* expression. Despite the aberrant activation of STAT3 signaling being associated with inflammatory processes and cancer, STAT3 is critical in maintaining cellular homeostasis rather than playing a singular role in acute phase responses [47]. Conversely, *PG* inhibited the activation of STAT3 on primary fibroblasts, which could be mediated by the reduced levels of IL-6. Several reports highlight the role of IL-6 trans-signaling and activation of the STAT3 pathway in mediating an exaggerated activation of fibroblasts leading to their transformation into cancer-associated fibroblasts [48]. The strong modulation of this pathway by polyphenols further enforces their beneficial effects not only in promoting the recovery of mucosal damage but also in preventing tumor development.

Based on these results, the consumption of *PG* promotes epithelium regeneration not only by acting directly on epithelial cells but also indirectly leading to the release of fibroblast-derived factors, which, in turn, amplify intestinal repair.

## 5. Conclusions

In summary, the present study shows how *PG* may exert beneficial effects on the recovery of mucosal damage in experimental models of IBD. If these results are confirmed from clinical studies in IBD patients, the consumption of *Punica granatum* extract may act as an adjuvant approach to maintain remission and reduce flare ups of the disease.

## Figures and Tables

**Figure 1 nutrients-15-01771-f001:**
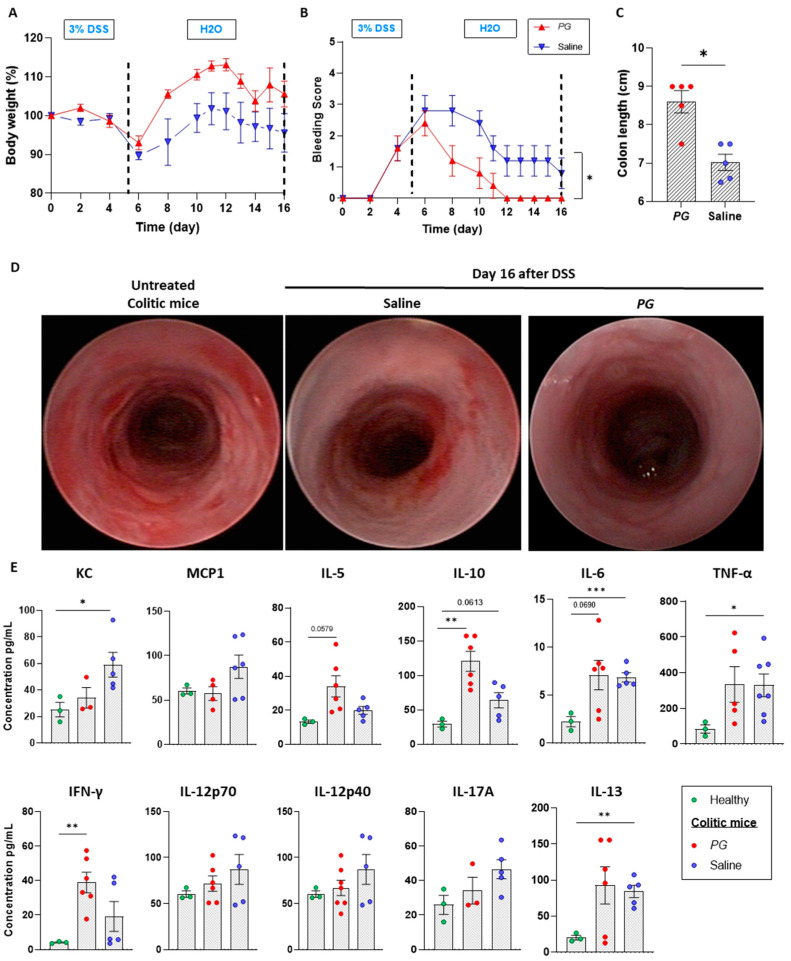
*PG* accelerated the recovery phase of acute colitis, promoting the mucosal epithelial barrier repair. (**A**) Body weight and (**B**) bleeding score were evaluated in the DSS-induced colitis model during the acute phase induced by 3% DSS administered ad libitum for 5 days and during the recovery phase replacing DSS treatment with regular water and treating mice with *PG* or saline. (**C**) Colon length (cm) was measured in *PG*- and saline-treated mice. (**D**) Intestinal wall damage of untreated colitic mice and of *PG*- and saline-treated mice were monitored by endoscopic evaluation. (**E**) Multiplex bead-based assay was performed on healthy and *PG*- and saline-treated mice. Levels (*PG*/mL) of KC, MCP1, IL-5, IL-10, IL-6, TNF-α, INF-γ, IL12p70, IL12p40, IL-17, and IL13 were evaluated. Data are presented as mean ± SEM. Unpaired *t*-test. * *p* < 0.05, ** *p* < 0.01, *** *p* < 0.001. *n* = 6 mice/*PG* group, *n* = 5 mice/saline group, *n* = 3 mice/healthy group.

**Figure 2 nutrients-15-01771-f002:**
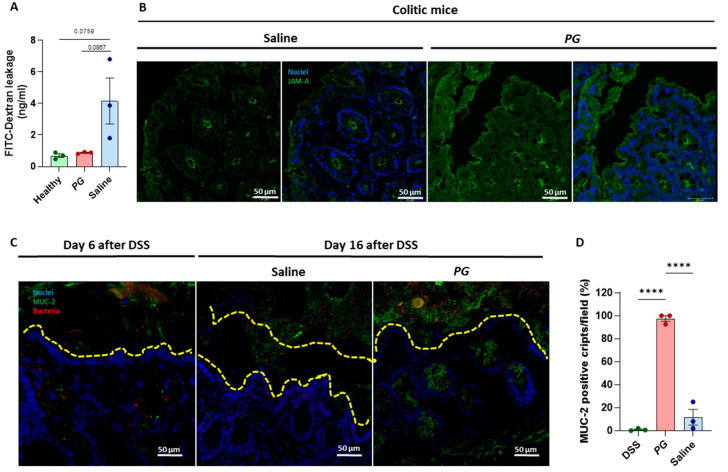
*PG* restored the intestinal barrier in vivo with the formation of a mucus layer. (**A**) Intestinal permeability was evaluated through FITC-dextran assay in healthy and *PG* or saline-treated mice, quantifying FITC-dextran levels in murine sera. (**B**) Representative images of immunofluorescence staining of JAM-A (green) in the epithelial layer and crypts of colitic mice during the recovery phase subjected to saline or *PG*. (**C**) Fluorescence in situ hybridization for bacterial rRNA (red) and immunofluorescence staining of MUC-2 (green); and (D) quantification of MUC-2 positive cripts for field (40X)was performed in the colonic mucosa of colitic mice before and after the recovery phase with the administration of saline or *PG*. (DAPI = nuclei in blue. Scale bar = 50 µm. Data are presented as mean ± SEM. One-way ANOVA test. **** *p* < 0.0001. *n* = 3 mice/group.

**Figure 3 nutrients-15-01771-f003:**
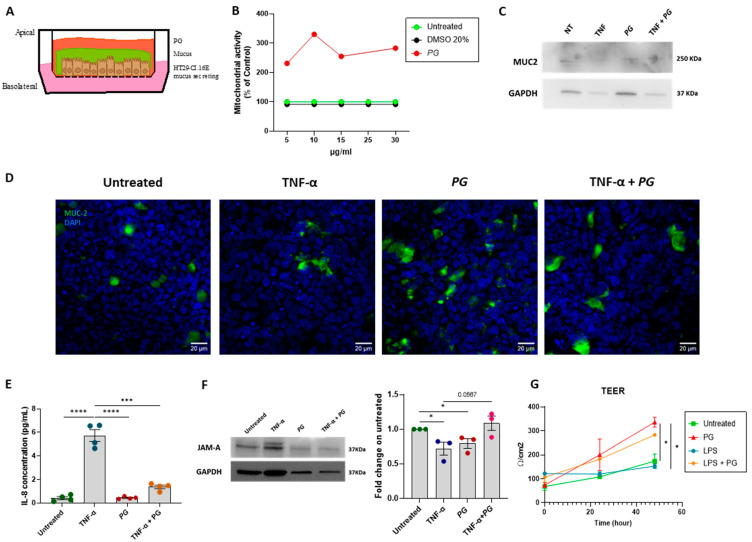
*PG* enhanced the epithelial barrier integrity in vitro, modulating the mucus layer and junctional adhesion protein production. (**A**) Schematic representation of in vitro experimental plan, in which the mucus secreting cells, HT29C1.16E, after apical and basolateral polarization on transwell supports, were stimulated with *PG*. (**B**) MTT assay was performed on HT29C1.16E cells after stimulation with *PG* at different concentrations (5, 10, 15, 25, 30 µg/mL) for 48 h. HT29C1.16E untreated or stimulated with 20% DMSO was used as the control. (**C**) Representative Western blot of MUC2 and GAPDH as the control in HT29C1.16E untreated and treated for 48 h with TNF-α, *PG*, or TNF-α + *PG*. (**D**) Representative immunofluorescence of MUC-2 (green) in HT29C1.16E treated for 48h ± TNF-α, *PG*, or TNF-α + *PG*. DAPI = nuclei in blue. Scale bar = 20 µm. (**E**) IL-8 (*PG*/mL) levels in Caco-2 cells treated for 48h ± TNF-α, *PG*, or TNF-α + *PG*. *n* = 4/group. (**F**) Representative Western blot of JAM-A and GAPDH as housekeeping protein in Caco-2 cells treated for 48h ± TNF-α, *PG* or TNF-α + *PG*. JAM-A quantification relative to GAPDH was expressed as fold change in the untreated group. (**G**) Transepithelial electrical resistance (TEER)(Ώ/cm^2^) was performed on Caco-2 cells treated ± *PG* or *PG* + LPS 24 and 48 h after stimulation. Data are presented as mean ± SEM. Unpaired *t*-test. * *p* < 0.05, *** *p* < 0.001, **** *p*<0.0001 *n* = 3/group.

**Figure 4 nutrients-15-01771-f004:**
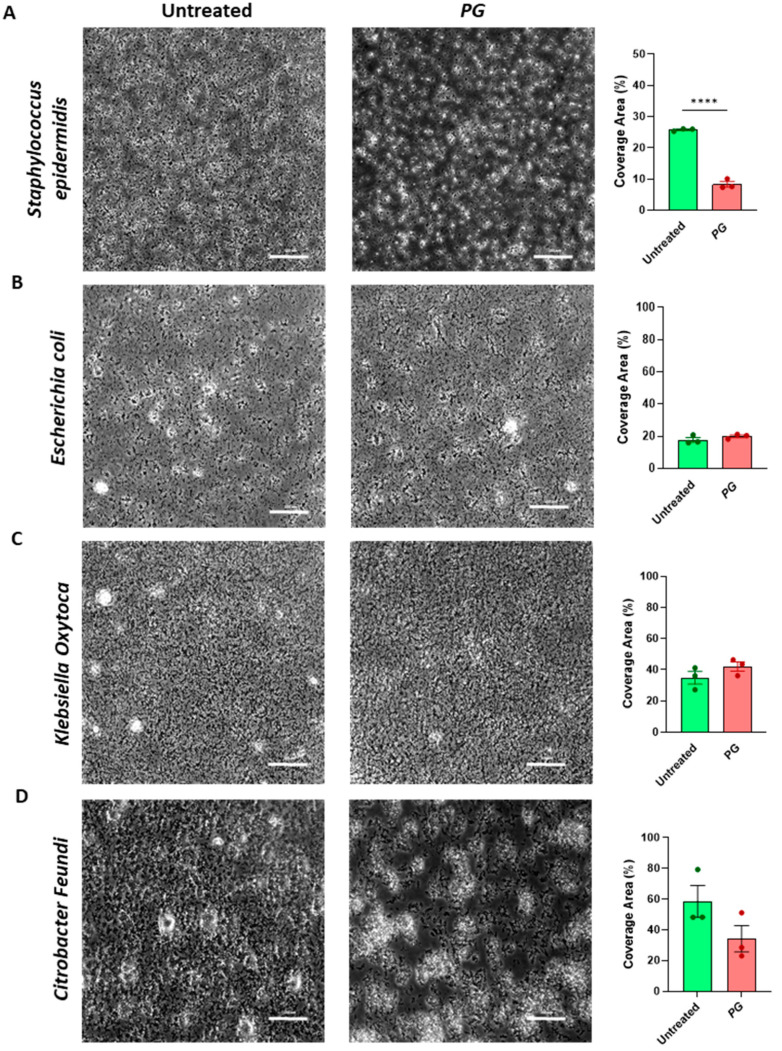
*PG*-mediated microbial biofilm. Biofilm formation was evaluated in (**A**) *Staphylococcus epidermidis*, (**B**) *Escherichia coli*, (**C**) *Klebsiella oxytoca*, and (**D**) *Citrobacter freundii* calculating the % of bacterial surface coverage from phase-contrast images. Data are presented as mean ± SEM. Unpaired *t*-test. **** *p* < 0.0001. *n* = 3/group.

**Figure 5 nutrients-15-01771-f005:**
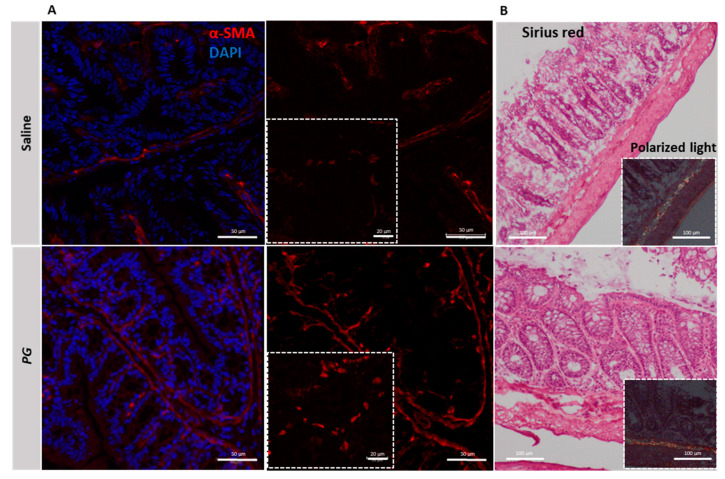
*PG*-activated fibroblasts during the recovery phase of acute colitis. (**A**) Representative immunofluorescence staining of α-SMA (red) in the colonic section of colitic mice in resolution phase treated with saline or *PG*. DAPI = nuclei in blue. Scale bar = 50 µm; Scale bar magnification = 20 µm. (**B**) Sirius red staining and polarized light microscopy showing the collagen deposition and the different collagen fibers (green, yellow, and red), respectively, in the colonic mucosa of colitic mice during the resolution phase subjected to saline or *PG* treatment. *n* = 3 mice/group. Scale bar = 100 µm.

**Figure 6 nutrients-15-01771-f006:**
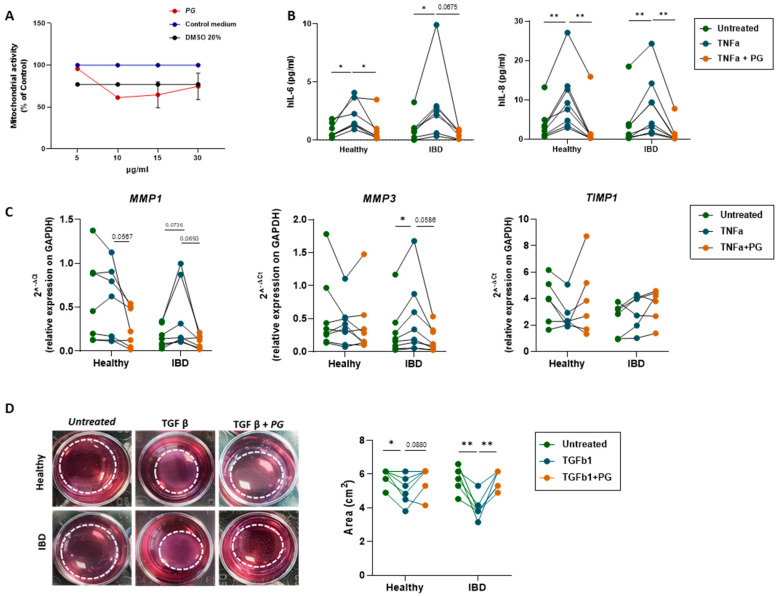
*PG* reverted fibroblast activation, decreasing inflammatory cytokines, MMP expression, and the contractile phenotype. (**A**) MTT assay was performed on primary intestinal fibroblasts isolated from noninflamed and inflamed areas of IBD patients after stimulation with *PG* at different concentration (5, 10, 15, 30 µg/mL) for 48h. Primary fibroblasts untreated or stimulated with 20% DMSO were used as control. (**B**) IL-6 and IL-8 levels (*PG*/mL) in healthy and IBD fibroblasts treated for 48h ± TNF-α or TNF-α + *PG*. (**C**) Quantitative Real-Time PCR analysis of *MMP1*, *MMP3,* and *TIMP1* mRNA expression in healthy and IBD fibroblasts stimulated ± TNF-α or TNF-α + *PG*. Gene expression was normalized to *GAPDH*. (**D**) Representative images of contraction assay performed on healthy and IBD fibroblasts stimulated ± TGF-β or TGF-β + *PG*. The collagen circles are highlighted by the dashed white line. The circle area (cm^2^) was measured in healthy and IBD fibroblasts stimulated ± TNF-α or TNF-α + *PG*. Data are presented as mean ± SEM. Paired *t*-test * *p* < 0.05, ** *p* < 0.01. *n* = 8 healthy/group, *n* = 8 IBD (5 CD and 3 UC)/group.

**Figure 7 nutrients-15-01771-f007:**
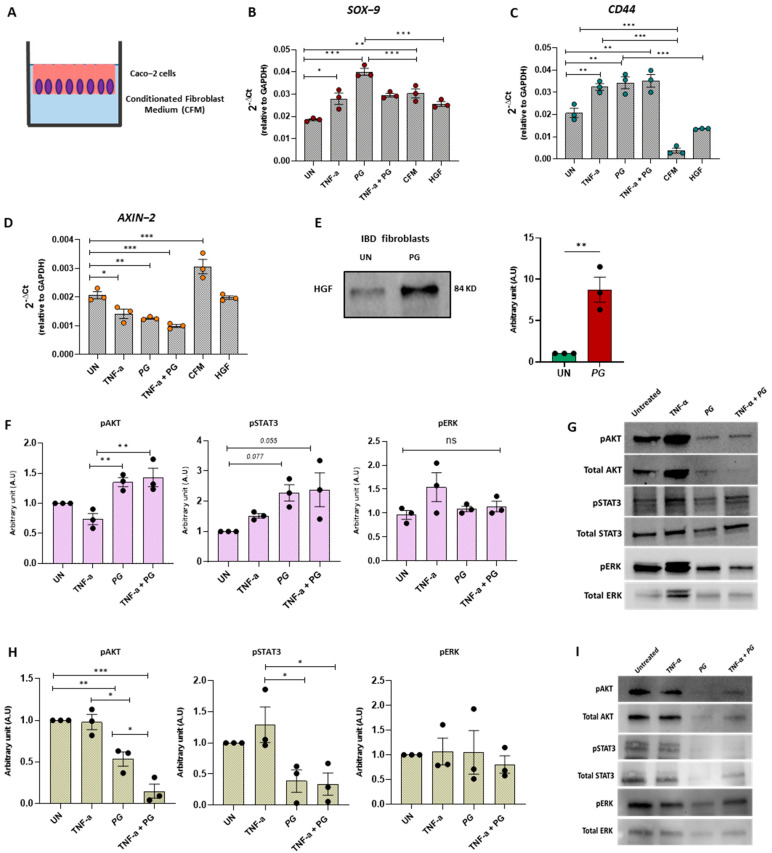
*PG* promoted intestinal epithelium repair by regulating fibroblast activity. (**A**) Experimental design of Caco−2 cells cultured for 24 h with conditioned medium (CFM) derived from IBD fibroblasts stimulated ± TNF-α, *PG*, or TNF-α + *PG*. (**B**−**D**) Quantitative Real-Time PCR analysis of *SOX*−*9*, *CD44*, and *AXIN*−*2* mRNA expression in Caco-2 cells treated ± TNF-α, *PG*, TNF-α + *PG*, CFM derived from fibroblasts stimulated with *PG* or HGF. Gene expression was normalized to *GAPDH*. (**E**) Western blot analysis of HGF in the supernatants of IBD fibroblasts ± *PG*. HGF was normalized on total proteins (**F**,**G**) Western blot analysis of pAKT, Total AKT, pSTAT3, Total STAT3, pERK, and Total ERK in Caco−2 cells ± TNF-α, *PG* or TNF-α + *PG*. (**H**,**I**) Western blot analysis of pAKT, Total AKT, pSTAT3, Total STAT3, pERK, and Total ERK in IBD fibroblasts ± TNF-α, *PG* or TNF-α + *PG*. Phosphorylated AKT, STAT3 and ERK were normalized on total AKT, STAT3, and ERK, respectively. Data are presented as mean ± SEM. One-way ANOVA test. * *p* < 0.05, ** *p* < 0.01, *** *p* < 0.001. *n* = 3/group.

**Table 1 nutrients-15-01771-t001:** Disease activity index (DAI) scores.

Score	Weight Loss (%)	Stool Consistency	Rectal Bleeding
0	<1	Normal	Negative
1	1–5		
2	5–10	Loose	Positive
3	10–15	Diarrhea	Slight bleeding
4	>15		Gross bleeding

**Table 2 nutrients-15-01771-t002:** List of the human primers utilized to perform RT-PCR.

Gene	Forward	Reverse
*GAPDH* *CD44*	5′ GGAGCGAGATCCCTCCAAAAT 3′ 5′ CTGCCGCTTTGCAGGTGTA 3′	5′ GGCTGTTGTCATACTTCTCATGG 3′ 5′ CATTGTGGGCAAGGTGCTATT 3′
*MMP1*	5′ CTCTGGAGTAATGTCACACCTCT 3′	5′ TGTTGGTCCACCTTTCATCTTC 3′
*MMP3*	5′ CTGGACTCCGACACTCTGGA 3′	5′ CAGGAAAGGTTCTGAAGTGACC 3′
*TIMP1*	5′ ACCACCTTATACCAGCGTTATGA 3′	5′ GGTGTAGACGAACCGGATGTC 3′
*AXIN2* *SOX9*	5′ CAACACCAGGCGGAACGAA 3′ 5′ AGCGAACGCACATCAAGAC 3′	5′ GCCCAATAAGGAGTGTAAGGACT 3′ 5′ CTGTAGGCGATCTGTTGGGG 3′

## Data Availability

Not applicable.

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
