# Peer review of "Pomegranate Extract Affects Gut Biofilm Forming Bacteria and Promotes Intestinal Mucosal Healing Regulating the Crosstalk between Epithelial Cells and Intestinal Fibroblasts"

_nutrients, 2023, doi:10.3390/nu15071771_

Round 1
Reviewer 1 Report
The present study investigates the effect of pomegranate extract on bacterial biofilm formation and promotion of mucosal wound healing. The experimental design is reasonable, the necessary experimental details are clearly explained, and the results are displayed accurately. Conclusion needs to be modified appropriately. Other comments are as follows.
1) If the data is expressed as mean ± SEM, it indicates that the data conforms to a normal distribution, so it should not be tested using Mann – Whitney test or Wilcox matched pairs.
2) Line 429: replace “ssignificant” by “significant”
3) Conclusions: This study is not a human intervention experiment and cannot conclude that it can improve IBD patients.
Author Response
Reviewer 1
The present study investigates the effect of pomegranate extract on bacterial biofilm formation and promotion of mucosal wound healing. The experimental design is reasonable, the necessary experimental details are clearly explained, and the results are displayed accurately. Conclusion needs to be modified appropriately. Other comments are as follows.
Q1. If the data is expressed as mean ± SEM, it indicates that the data conforms to a normal distribution, so it should not be tested using Mann – Whitney test or Wilcox matched pairs.
A1. We thank the Reviewer for this comment. We revised the statistical power applying a parametric analysis. The figures have been updated appropriately.
Q2. Line 429: replace “ssignificant” by “significant”
A2. The typo has been corrected
Q3. Conclusions: This study is not a human intervention experiment and cannot conclude that it can improve IBD patients.
A3. The conclusions have been revised as follows: In summary, the present study shows how PG may exert beneficial effects on the recovery of mucosal damage in experimental models of IBD. If these results are confirmed from clinical studies in IBD patients, the consumption of Punica granatum extract may act as an adjuvant approach to maintain remission and reduce flare-ups of the disease.

Reviewer 2 Report
In this manuscript, polyphenols suppress inflammatory bowel disease and show the possibility of suppressing recurrence by restoring the mucosa.It's interesting overall, but it might be easier to accept if the fluorescence image, which is the important data in the first half, is shown more clearly.Check out the small suggestions below.
1. The annotation on the upper left of the fluorescent immunostaining image in Fig.2 is unnecessary. And show the image on the right to be the merge.
2. The band in Fig3-C is faint, so please provide a clearer image if possible.
3. Show Fig3-D that the bottom image is a merge as well. In addition, it is necessary to increase the resolution of the image by, for example, increasing the magnification.
4. There seems to be no LPS-treated group in Fig3-G. It is recommended to add to powerfully show the effect of PG.
5. The left and right images in Fig5-A are paired, but they appear to be in slightly different positions. Please use images of the same position and indicate that it is a merge.
6. The caption for Table 2 is below the table. This caption should be shown above the table.
7. Due to the heavy use of abbreviations, I suggest adding an abbreviations glossary.
8. Is "PEG" on line 425 a mistake of "PG"? please confirm.
Author Response
Reviewer 2
In this manuscript, polyphenols suppress inflammatory bowel disease and show the possibility of suppressing recurrence by restoring the mucosa. It's interesting overall, but it might be easier to accept if the fluorescence image, which is the important data in the first half, is shown more clearly.Check out the small suggestions below.
Q1. The annotation on the upper left of the fluorescent immunostaining image in Fig.2 is unnecessary. And show the image on the right to be the merge.
A1. We revised the fluorescent immunostaining as requested.
Q2. The band in Fig3-C is faint, so please provide a clearer image if possible.
A2. We agree with the reviewer, we tested several different antibodies for this analysis, the only one highly specific was this, but the band was fair. All replicated are similar, therefore we decided to maintain this WB.
Q3. Show Fig3-D that the bottom image is a merge as well. In addition, it is necessary to increase the resolution of the image by, for example, increasing the magnification.
A3. We revised the images as requested
Q4. There seems to be no LPS-treated group in Fig3-G. It is recommended to add to powerfully show the effect of PG.
A4. We included the LPS group, as requested.
Q5. The left and right images in Fig5-A are paired, but they appear to be in slightly different positions. Please use images of the same position and indicate that it is a merge.
A5. We revised the images as requested
Q6. The caption for Table 2 is below the table. This caption should be shown above the table.
A6. We revised the caption of the table as requested.
Q7. Due to the heavy use of abbreviations, I suggest adding an abbreviations glossary.
A7. Thank you for this comment, we have included a glossary to make reading the manuscript easier.
Q8. Is "PEG" on line 425 a mistake of "PG"? please confirm.
A8. The typo has been corrected.
